# The influence of habitat use on harvest vulnerability of cow elk (*Cervus canadensis*)

**Maksim Sergeyev** [1]*, **Brock R. McMillan**[1], **Kent R. Hersey**[2], **Randy T. Larsen**[1]

**1** Department of Plant and Wildlife Sciences, Brigham Young University, Provo, UT, United States of America, **2** Utah Division of Wildlife Resources, Salt Lake City, UT, United States of America

* ecomaksimsergeyev@gmail.com

## Abstract

Pressure from hunting can alter the behavior and habitat selection of game species. During hunting periods, cervids such as elk (*Cervus canadensis*) typically select for areas further from roads and closer to tree cover, while altering the timing of their daily activities to avoid hunters. Our objective was to determine the habitat characteristics most influential in predicting harvest risk of elk. We captured 373 female elk between January 2015 and March 2017 in the Uinta-Wasatch-Cache National Forest and surrounding area of central Utah, USA. We determined habitat selection during the hunting season using a resource selection function (RSF) for 255 adult cow elk. Additionally, we used a generalized linear mixed model to evaluate risk of harvest based on habitat use within home ranges (3rd order selection) as well as the location of the home range on the landscape to evaluate vulnerability on a broader scale. Female elk selected for areas that reduced hunter access (rugged terrain, within tree cover, on private land). Age, elevation and distance to roads within a home range were most influential in predicting harvest risk (top model accounted for 36.2% of AIC weight). Elevation and distance to trees were most influential in predicting risk when evaluating the location of the home range (top model accounted for 42.1% of AIC weight). Vulnerability to harvest was associated with proximity to roads. Additionally, survival in our landscape decreased with age of female elk.

## Introduction

Selection of resources and habitats is a driving force influencing animal population [1]. As such, a thorough understanding of the factors that influence habitat selection is vital for proper management and conservation of a species [2]. Because resources are not uniformly available across the landscape, organisms select the most beneficial habitats [3]. Selection occurs at multiple scales and has been categorized into specific orders of selection [4]. The broadest of these scales, first order selection, describes selection of a geographic range, while second order narrows the selection further to local sites [5]. Third order selection describes usage patterns of local areas and finally, fourth order selection can describe selection at finer scales (e.g., foraging sites). Selection of habitats may be influenced by quality of forage, risk of predation, competition, energy trade-offs, or anthropogenic influences like development, outdoor recreation, and hunting [6–8].

**Data Availability Statement:** Data is in a GitHub repository, the link is as follows: https://github.com/MaksimSergeyev/HarvestVulnerabilityElk.

**Funding:** This work was supported by the Rocky Mountain Elk Foundation, Sportsmen for Fish and

Wildlife, Utah Division of Wildlife Resources, and Brigham Young University (Grant #R0402272). Brigham Young University received grant number #R0402272 as part of a collaborative effort with the Utah Division of Wildlife Resources to monitor elk movement throughout the state of Utah.

**Competing interests:** The authors have declared that no competing interests exist.

Pressure from hunting (additional disturbance, increased risk of mortality) can influence behavior and habitat selection of game species. During hunting periods, game species often shift habitat use away from areas with optimal resource quality towards areas offering greater security [9]. For example, black bears (*Ursus americanus*) and wolves (*Canis lupus*) shifted habitat use towards less accessible areas, further from roads [6]. White-tailed deer (*Odocoileus virginianus*) altered their habitat use and timing of daily activity to avoid hunters [10]. Hunting led to reduced intraspecific competition, decreased mating opportunities, and increased group size in red deer (*Cervus elaphus*) and Dall sheep (*Ovis dalli*), likely due to the removal of dominant individuals [10, 11]. Understanding the effects of harvest and anthropogenic activities on behavior, resource selection, and population dynamics is fundamental to conservation.

Rocky Mountain elk (*Cervus canadensis*), a big game species in North America respond to hunting pressure suggesting that hunters may influence elk population dynamics beyond the direct effects of harvest-related mortality. During the hunting season, elk select for areas further from roads and often use private land as refuge [12–15]. Daily movement rates increase and elk expend additional energy avoiding hunters [16, 17]. Additionally, flight distances of elk increase during hunting periods, while group sizes decrease, suggesting elk are aware of the increased risk of mortality [17, 18]. Not only can hunting pressure influence distribution of elk, the distribution of elk on the landscape may influence susceptibility to harvest. Vulnerability of elk to harvest is likely influenced by hunter efficiency, habitat selection by elk, and detectability of elk [19]. Detectability of elk can vary with time of day, cover type, and presence of snow. It may also decrease with age as older individuals become familiar with annual hunting pressure [20].

As elk age, they may learn to avoid hunters by reducing use of high-risk areas [20, 21]. Bull elk had more pronounced responses to hunting pressure than cows and mature bulls exhibited greater flight distances than younger bulls, consistent with higher rates of harvest for mature bulls [18]. Older cow elk reduced movement rates during the hunting period and increased use of rugged terrain [20]. Further, the same study showed that cows over the age of 9 or 10 were almost invulnerable to harvest by hunters. As long-lived, gregarious animals, elk may learn to avoid hunters by altering habitat use.

The risk of harvest for game animals is likely influenced by a multitude of factors, including selection of habitat during the hunting season. Our objectives were to determine the habitat characteristics most influential in predicting harvest risk of elk. We expected risk of harvest to be correlated with hunter accessibility and that elk in rugged, less-accessible areas would be at reduced risk. Further, we predicted older elk would show reduced use of high-risk areas. We treated the rifle season and archery season as one long hunting season. While differences may have existed between the two seasons, the general patterns and predictions associated with harvest risk shouldn't change. Elk that selected habitats closer to roads and in less rugged terrain, for example, were expected to be at increased risk. While we were unable to evaluate all factors that potentially influenced the habitat use of elk during the hunting season, we compared patterns of use during the hunting season and examined differences in use between hours of risk (daytime) and hours when hunting was not allowed (nighttime) to provide an overall understanding of how habitat use changes during hours of harvest risk. Identifying the factors associated with harvest risk of elk can increase knowledge of population dynamics, advance understanding of the responses of game species to hunters, and provide additional insight into age structure of the population, thereby improving management.

## Methods

### Study area

This study was part of a larger study on the movement patterns and habitat use of elk, conducted in the Wasatch Mountains and surrounding areas of central Utah, USA west of Salt

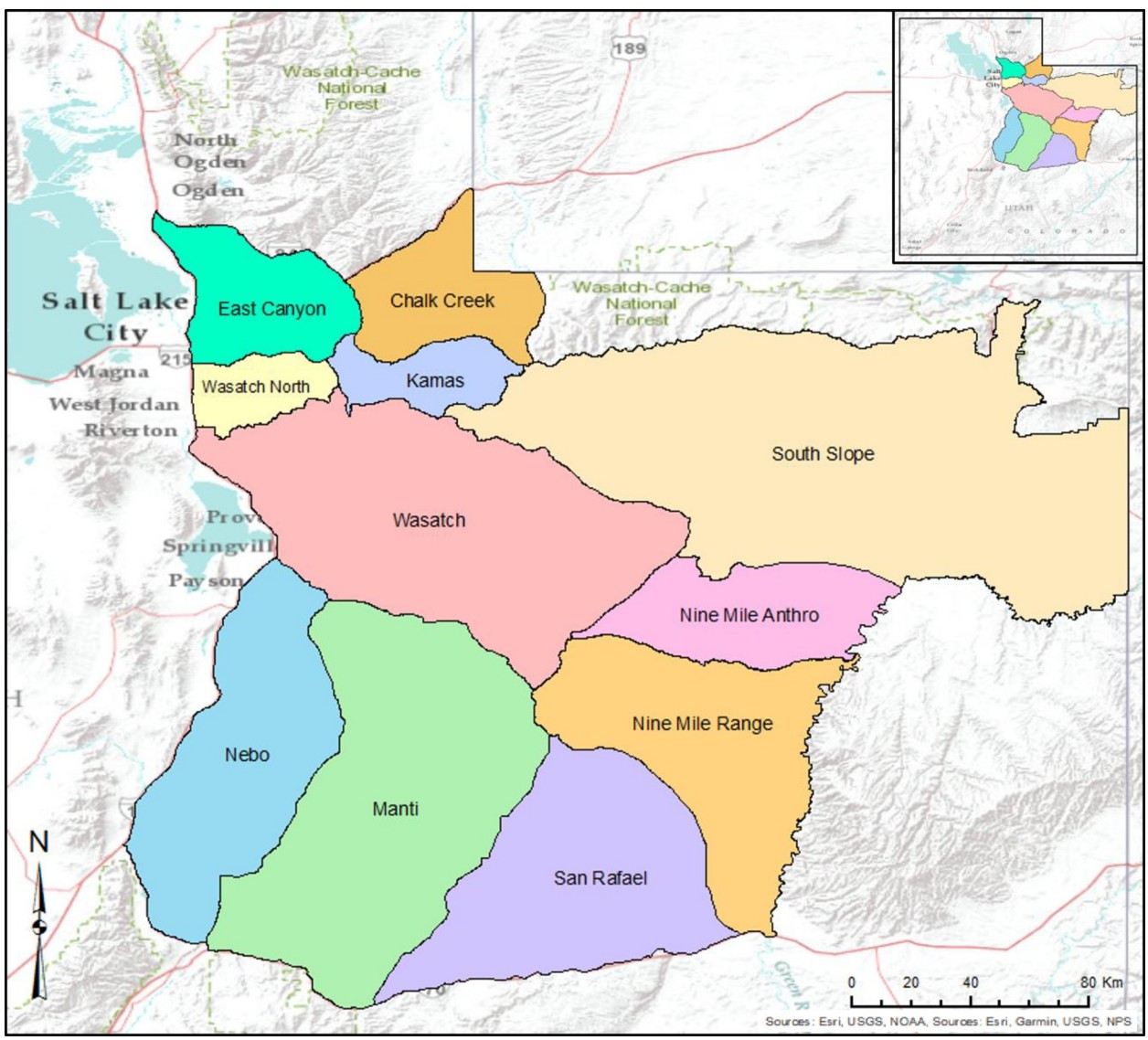

**Fig 1. Study of harvest vulnerability of elk was conducted in the Wasatch and surrounding management units of central Utah, U.S.A.** Colored polygons denote the separate management units in the area.

Lake City (Fig 1). The Wasatch Mountains are the southwestern portion of the Rocky Mountains extending approximately 400 kilometers [22], and have steep, rugged slopes from past glaciation events [23]. The Wasatch range is formed from dolomite and limestone [24]. This area receives an approximately 40 centimeters of annual precipitation, which varies with elevation [25]. Base elevation in the region is approximately 1370 meters and Mount Nebo, at 3620 meters, is the highest point along the range, as well as other prominent peaks such as Mount Timpanogos and Mount Olympus [26, 27]. The species composition of plant communities varies with elevation and have been described across distinct elevational zones [28]. Below 1980 meters, the Upper Sonoran Zone, is dominated by sagebrush (*Artemisia* spp) and Mexican cliffrose (*Purshia stansburyana*). Elevations between 1981–2440 meters, the Transition Zone, are populated by brush species like Gambel oak (*Quercus gambelii*) and curl-leaf mountain mahogany (*Cercocarpus ledifolis*; [29]). Above the Transition Zone is the Canadian Zone,

from 2440–2900 meters, which is characterized by aspen (*Populus tremuloides*) and white fir (*Abies concolor*), then followed by the Hudsonian Zone, characterized by subalpine fir (*Abies lasiocarpa*) and Engelmann spruce (*Picea engelmannii*). Finally, the Arctic-Alpine Life Zone, above 3200 meters, is populated by primrose (*Oenothera* spp) and alpine moss. Predators of elk in this region are limited; no wolves or grizzly bears (*Ursus arctos*) occur and predation from mountain lions (*Puma concolor*) is minimal, typically restricted to calves or older, weakened individuals.

**Elk hunting in Western US.** Because hunting is the primary tool used to manage populations, the goal in most regions of the western Unites States is to provide opportunity for hunters while managing populations. To attain these management goals, there are often different game management units with different goals ranging from primarily opportunity (increased opportunity to harvest any animal) to primarily "quality (i.e., increased likelihood of harvesting a mature male)". Different seasons (e.g., archery only, muzzleloader only, and any weapon) for different portions of the population (females only, males only, or either sex) and the various seasons generally run from August through January. In our study area, archery-only seasons typically start in mid-August and run through mid-late September. Any weapon hunts (most hunters electing to hunt with a rifle) start in October and end in January. Of course, differences among weapons types and seasons affect the likelihood of successfully harvesting an animal [30]. Hunter's using archery tackle are typically less successful than those using firearms. Likewise, it is hypothesized that hunters pursuing females are typically more successful than hunters pursuing males unless it is a unit managed for high "quality" as these populations generally have a relatively high proportion of males.

Methods of take primarily consist of ambush, still-hunting, or spot and stalk methods. For the ambush method, hunters generally sit in a location of vantage (hill, tree stand, etc.) in areas known to have elk or along travel paths and wait for an individual to travel into the open within their effective hunting range. The still-hunting method consists of the hunter moving quietly and very slowly, with regular stops, through elk habitat in an attempt to spot an animal before the animal is aware of the hunter's presence. The spot and stalk method is when a hunter searches from a vantage with a spotting scope or other optics in an attempt to locate game, often at a great distance (up to several km away). Once the game is located, the hunter plans an approach (i.e., stalk) to get within effective hunting range without detection. There are a few other less common methods (e.g., small-scale game drives by groups of hunters, driving roads while looking for elk, etc.).

## Elk capture

We captured elk via helicopter net-gunning from January 2015 through March 2017 [31]. Individuals were restrained using hobble straps and fit with a blindfold, no sedation or euthanasia was involved. All capture and handling of wildlife was approved by the Brigham Young University Institutional Animal Care and Use Committee (permit #150112).We collected body measurements, blood, and fecal samples for each elk, as well as an estimate of body condition [32] and age based on dental wear. Approximately 10–16 mLs of blood were collected from each individual from either the neck or the leg and subsequently used for mineral analysis. Aging by tooth wear is not always accurate, however, post-hoc analyses of teeth from deceased elk via cementum analysis found that over 80% of individuals were accurately aged to within one year. We measured loin muscle thickness and rump fat using ultrasonography. Body mass and ingesta free body fat were calculated for each individual [33]. Captured individuals were then fitted with VHF radio and GPS collars before being released. In order to balance frequency of data collection and longevity of the collars, we programmed collars to collect a

GPS location every 13 hours. Mortality warnings were triggered by a lack of animal movement after 8 hours. When we received a mortality warning, we attempted to locate the deceased animal and determine cause of death within 48 hours.

## Analysis

**Evaluating habitat use during hunting season.** We calculated separate home ranges for each elk and each hunting season. We restricted our analysis to locations collected during the hunting season to limit any additional effects from seasonal changes. We created 95% minimum convex polygons (MCPs) using locations during the hunting season [34–36]. Opening day varied a little bit each year, but always occurred in mid-August; conclusion of the fall hunting season did not vary and ended on January 31st each year. A separate MCP was created for each hunting season, such that if an elk lived through all three years examined, three separate MCPs were created. We excluded animals with less than 50 locations during the hunting season to avoid biased estimates of home ranges [37, 38]. Although prior studies have raised questions on the legitimacy of MCPs to estimate home ranges, we used this method as a boundary to evaluate selection of each individual, as opposed to estimating size of the home range. Additionally, habitat occupied by elk did not include any uninhabitable areas that could have resulted in inaccurate MCPs. As a result, we considered these boundaries adequate to evaluate selection within the area used by each individual. We analyzed selection preferences during the hunting season using a resource selection function (RSF) to provide an understanding of habitat selection and distribution of individuals [39]. Based on known locations of use from collared individuals, relative probabilities of use can be estimated with an RSF [40, 41]. Within each MCP, we compared an equal number of used locations to random locations, after confirming that this number was sufficient to capture availability by looking at whether or not 95 percent confidence intervals from random locations spanned the mean from all pixels within a home range. We acquired habitat variables at 30-m resolution from the Utah Automated Geographic Reference Center (AGRC). Habitat variables included a 30-meter digital elevation model (DEM), land ownership, a layer of roads in the area, and dominant vegetation. As forest roads are frequently used during hunting season, we combined motorized roads and forest roads into a single layer to compute distance to roads. Ruggedness, slope, and aspect layers were created from the DEM using ArcGIS. Distance to trees, roads, and private land were computed in ArcGIS based on the rasters acquired from Utah AGRC. We included distance to private land as a variable as private land is often used as a means of refuge from hunting pressure by game species [12, 15]. While some landowners do allow hunting, this land is not available to the general public and, as such, typically experiences reduced or no hunting pressure. Other factors may be confounding the habitat selection of elk, such as seasonal changes, presence of predators, or varying pressure from hunters throughout the season or between weapon types. However, all samples (whether from harvested elk or elk that survived the hunting season) were treated identically and analyzed in the same manner limiting potential bias.

To examine habitat use during the hunting season, we evaluated 27 candidate models of habitat selection using an AICc model selection process for logistic regression models in program R [42, 43]. Variables were screened for collinearity in program R. We categorized each location as occurring during hunting hours, when elk are susceptible to risk, or outside of hunting hours, when risk of harvest is absent. Legal hunting hours started 30 minutes prior to sunrise and ended 30 minutes after sunset. We used these same time intervals to classify locations as occurring day or night. To examine differences in selection between day (i.e., when elk are susceptible to harvest) and night, we used interaction terms between habitat variables and a binary variable to denote the time as either day (1) or night (0). We formulated animal ID as

a random intercept to account for repeated measures and dependence of the locations [44]. We validated our top model using k-fold cross validation with k = 5 and computed a Spearman's rank correlation.

**Evaluating harvest vulnerability based on 3rd order habitat use.** We evaluated harvest vulnerability of elk based on habitat use at two scales: habitat use within home ranges [19] and at a broader scale based on the overall location of the home range on the landscape using the centroid of each home range [36]. We modeled risk of harvest by hunters using logistic regression with 1 corresponding to survival and 0 representing harvest [19]. We included variables for distance to roads, aspect, elevation, slope, terrain ruggedness, distance to tree cover, and distance to private land [14, 19]. Variables were screened for collinearity in program R. We evaluated vulnerability to harvest based on 3rd order selection by averaging data from all locations within the home range and considered each hunting season from every elk as an individual observation [45], resulting in a data set with each row containing the average value of each variable for one elk during one season. We excluded locations that occurred outside of hunting hours, as there was no risk of harvest mortality during these hours. We used linear mixed-effects regression models to examine habitat characteristics as fixed effects and again incorporated animal ID as a random intercept to account for dependence between repeated observations for a single individual. We evaluated 20 candidate models of harvest vulnerability using an AICc selection process in program R [42, 43]. Some of these models included age to determine if older elk were less susceptible to harvest than younger elk.

**Evaluating harvest risk based on location of home range.** Additionally, we evaluated harvest risk based on the location of the home range on the broader landscape using the centroid of each home range [36]. We chose to use the center of the home range to examine how risk of harvest may change based on the overall position of the home range on the broader landscape. As such, the centroid can be used to evaluate this broader geographic location. We obtained measurements of the aforementioned habitat characteristics for the centroid of each home range. We evaluated the same set of 20 candidate models to compare influential habitat characteristics between the two scales. Using the top model, we developed a map of risk of hunter harvest across the study area [46].

## Results

Between January of 2015 and March of 2017, we captured and collared 373 female elk. We restricted the analysis to locations during the hunting season and removed any elk with less than 50 locations, at which point 255 animals remained. We created separate home ranges for each hunting season during which an animal had locations, totaling 358 home ranges. In total, 80 animals were harvested by hunters throughout the 3-year period, 58 of which were included in this analysis based on the criteria described above.

Out of 27 candidate models of habitat use, the top model accounted for 83.6% of the weight compared to 16.4% for the second most supported model (Table 1; Spearman's rank $\rho$ = 1, p = 0.0167). Habitat use of elk during the hunting season was influenced by aspect, elevation, ruggedness, slope, and distance to private land, trees, roads, day vs night, and an interaction between time of day and ruggedness, distance to private land, and distance to trees. According to the interaction terms in the model, elk selected for rugged terrain, closer to private land and tree cover during the day compared to nighttime (Table 2). Overall, elk selected for areas that were high in elevation and far from roads and tree cover. Steep slopes and rugged terrain were associated with decreased probability of use.

We determined habitat factors that had the greatest support for predicting risk of harvest and found differing results between the two scales examined. Within each animal's home

**Table 1. AICc model selection results for 27 candidate models of habitat use.**

| | d.f. | AICc | ΔAICc | Weight |
|---|---|---|---|---|
| Aspect + Elevation + Day*Ruggedness + Slope + DistTrees + DistRoads + Day + DistPriv + Day*DistPriv + Day*DistTrees | 13 | 191845.2 | 0.00 | 0.836 |
| Aspect + Elevation + Ruggedness + Slope + DistTrees + DistRoads + Day + DistPriv + Day*DistPriv + Day*DistTrees | 12 | 191848.5 | 3.26 | 0.164 |
| Aspect + Elevation + Ruggedness + Slope + DistTrees + DistRoads + Day + DistPriv + Day*DistPriv | 11 | 192073.3 | 228.10 | 0.000 |
| Aspect + Elevation + Ruggedness + Slope + DistTrees + DistRoads + DistPriv+ Day + Day*DistTrees | 11 | 192214.2 | 368.97 | 0.000 |
| Null | 1 | 193528.7 | 1683.48 | 0.000 |

Top model included aspect, elevation, ruggedness, slope, distance to trees, distance to roads, and distance to trees, accounting for 83.6% of the total weight. The variable Day represents a binary variable identifying a location as occurring within hours of hunting risk (Day, 1) or outside of hunting hours (0). We included Animal ID as a random effect in every model. The top five models based on AICc are included in the table.

range, harvest vulnerability was most influenced by distance to roads, elevation, and age of the animal (top model accounted for 36.2% of the weight, Table 3). According to our top model, harvest risk increased with proximity to roads (p = 0.056, Table 4, Fig 2). Additionally, probability of survival during hunting season was lower at higher elevations (Fig 3) and for older animals (Fig 4). Interactions terms between age and distance to roads, distance to trees, distance to private land, and elevation were not supported in the models.

Based on overall location of the home range on the landscape, vulnerability to harvest was most influenced by elevation and distance to trees (top model accounted for 42.1% of the weight, Table 5). The top model included an interaction between elevation and distance to trees (p = 0.028, Table 6) suggesting that at higher elevations, distance to trees became more influential in predicting harvest risk. At higher elevations, increasing distance to tree cover was positively associated with survival. We used the top model based on home range centroids to create a heatmap of harvest vulnerability across the study area (Fig 5) to illustrate high-risk areas. Our results predicted high vulnerability in the northwest (Currant Creek/Wasatch front) and southwest portions (Nebo Mountains) of the study area, as well as throughout the Uinta Mountains near the center of the study site. Additionally, we predict low vulnerability in the southeastern portion (Uinta basin).

**Table 2. Output from top model (based on AICc) of habitat selection of elk during the hunting season.**

| | Estimate | Std. Error | p–Value |
|---|---|---|---|
| Intercept | -0.0045 | 0.0073 | 0.533 |
| Aspect | 0.0634 | 0.0054 | < 0.001 |
| Elevation | 0.0394 | 0.0066 | < 0.001 |
| Day | 0.0133 | 0.0109 | 0.221 |
| Ruggedness | -0.0448 | 0.0077 | < 0.001 |
| Slope | -0.1368 | 0.0062 | < 0.001 |
| DistTrees | 0.0992 | 0.0084 | < 0.001 |
| DistRoads | 0.0881 | 0.0059 | < 0.001 |
| DistPriv | 0.0078 | 0.0080 | 0.331 |
| Day*Ruggedness | 0.0251 | 0.0109 | 0.022 |
| Day*DistPriv | -0.2163 | 0.0113 | < 0.001 |
| Day*DistTrees | -0.1663 | 0.0111 | < 0.001 |

The variable Day represents a binary variable identifying a location as occurring within hours of hunting risk (Day, 1) or outside of hunting hours (0).

**Table 3. AICc model selection results for 20 candidate models of survival based on habitat use.**

|  | d.f. | AICc | ΔAICc | Weight |
|---|---|---|---|---|
| Age + DistRoads + Elevation | 5 | 283.7 | 0.00 | 0.369 |
| DistRoads + Elevation + Age + DistPriv | 6 | 285.0 | 1.37 | 0.186 |
| Age + Elevation + DistRoads + Ruggedness | 6 | 285.7 | 2.07 | 0.131 |
| Age + Elevation | 4 | 286.1 | 2.45 | 0.108 |
| Age + Elevation + Age*Elevation | 5 | 287.4 | 3.73 | 0.057 |
| Null | 1 | 295.1 | 11.48 | 0.001 |

We included Animal ID as a random effect in every model. Models with greater than five percent of the cumulative weight are listed below. Top model included age, distance to roads, and elevation, accounting for 36.2% of the total weight. The top five models based on AICc are included in the table.

## Discussion

Overall, elk selected for areas at high elevations, far from roads and further from tree cover. Elk altered habitat selection during hunting hours, selecting for areas that limited hunter access. Specifically, elk selected for rugged terrain, tree cover and private land when risk of mortality was greater. Additionally, we found preference for flatter, less rugged terrain. Our study incorporated similar variables as prior studies in assessing habitat selection by elk such as vegetation and cover, road density, land ownership, topographical complexity, and various measures of hunter effort or access [19, 47, 48]. Based on our top model, elk altered their selection preferences during hunting hours, increasing use of areas with limited hunter access (rugged terrain, close to private land and tree cover), supporting our predictions and suggesting elk altered their habitat use in response to heightened risk [49]. We also found preference for flatter, less rugged areas further from tree cover, contrary to our expectations, however during the winter elk may select flatter grasslands for forage [9], possibly explaining the use of flatter, open areas. During the hunting season, elk selected for rugged areas with lower road density, closer to tree cover, consistent with other populations of elk [19, 45]. Additionally, we found a preference for private land, consistent with prior studies of hunted populations of elk [12, 14, 15]. Our results support previous findings of elk selecting for areas with limited hunter access during periods of heightened risk of harvest [50].

Within an animal's home range, harvest vulnerability was best predicted by distance to roads, age of the individual, and elevation. Elk had increased survival further from roads. Survival decreased with increasing elevation. This was likely due to public land generally occurring at higher elevations than private land within our study area; as hunting primarily occurred on public land, this may explain the decreased survival at higher elevations. Vulnerability of elk to harvest is often correlated with road density or proximity to roads [19, 45, 47, 51]. Our results support the idea that harvest risk increases with proximity to roads [50]. Additionally, survival decreased with age [52], and none of our models with age and habitat variable interactions were among the top supported models. This finding differs from previous work where elk have been shown to learn to avoid hunters with age. Older females reduced

**Table 4. Output from top model (based on AICc) of survival of elk based on habitat use.**

|  | Estimate | Std. Error | p–Value |
|---|---|---|---|
| Intercept | 1.932 | 0.171 | < 0.001 |
| Age | -0.113 | 0.151 | 0.453 |
| DistRoads | 0.414 | 0.222 | 0.0625 |
| Elevation | -0.311 | 0.176 | 0.0764 |

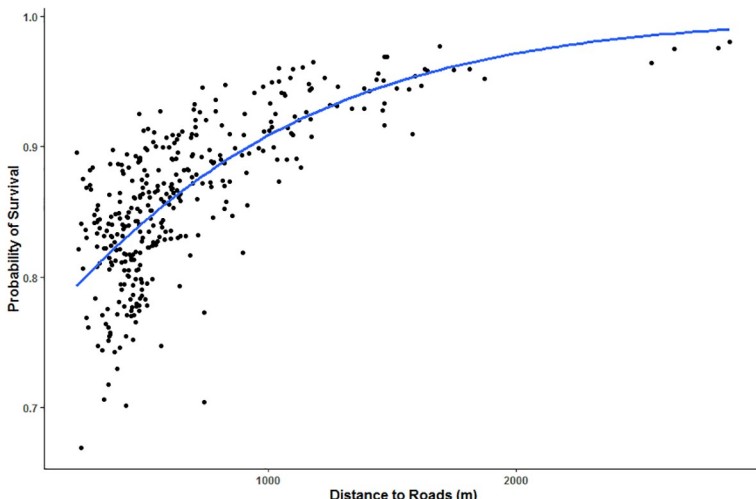

**Fig 2. Predictive model of harvest vulnerability of elk in central Utah, U.S.A., as a function of increasing distance to roads, according to the top model from AICc selection.** Top model included age, distance to roads, and elevation.

movement rates in the presence of hunting pressure and, as a result, had higher survival [53]. Older cow elk also increased use of rugged terrain closer to roads [20]. The specific reasons we did not detect learning in our study and why older elk were more likely to be harvested are unclear. In our study, we experienced relatively high harvest rates (23 percent; 58/255) that may have overcome evidence of learning. Alternatively, learning may have occurred, but because we only monitored elk for 3 seasons and many of our animals were not caught until the 2nd or 3rd year it simply wasn't detected. Thurfjell et al. [20], for example, monitored elk for up to 5 years. Further evaluation of learning by elk in response to hunting is warranted.

Based on the centroid of the home range, risk of harvest was best predicted by distance to trees, elevation, and an interaction between the two. Probability of survival was higher with increasing distance to trees at high elevations, somewhat contradictory to our expectations.

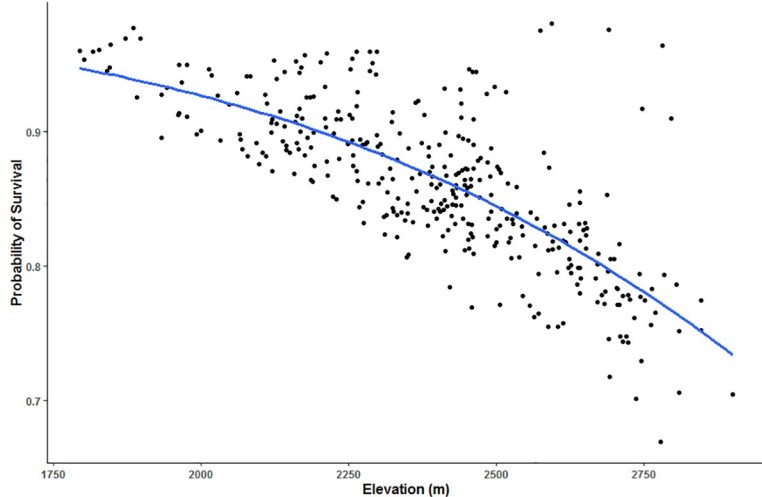

**Fig 3. Predictive model of harvest vulnerability of elk in central Utah, U.S.A., as a function of elevation, according to the top model from AICc selection.** Top model included age, distance to roads, and elevation.

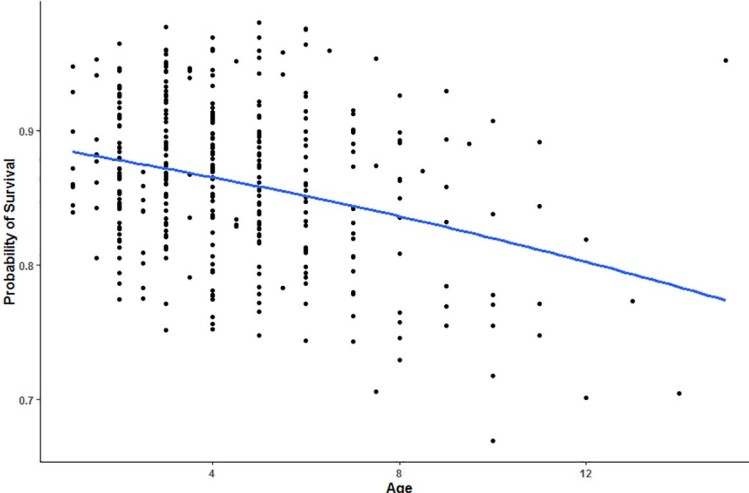

**Fig 4. Predictive model of harvest vulnerability of elk in central Utah, U.S.A., as a function of increasing age.**
Values are derived from top model according to AICc selection. Top model included age, distance to roads, and
elevation.

However, overlap between elk and hunters was highest in forested areas and lower in uncov-
ered areas [51], possibly explaining why we found lower harvest risk away from forest cover.
Additionally, elk decreased use of forested areas during the hunting season [20, 47], consistent
with our results that survival increased as distance to trees increased.

Elk altered habitat use during hunting hours, increasing use of areas with limited hunter
access (rugged terrain, within tree cover and closer to private land). Additionally, elk selected
for areas far from roads and high in elevation. On a broader scale, vulnerability to harvest was
influenced by elevation and distance to trees. Age, elevation and distance to roads were the
best predictors of harvest risk based on habitat use within the home range. Much is known

**Table 5. AICc model selection results for 20 candidate models of survival based on overall location of the home range on the landscape.**

|  | d.f. | AICc | ΔAICc | Weight |
|---|---|---|---|---|
| Elevation + DistTrees + Elevation*DistTrees | 5 | 239.0 | 0.00 | 0.421 |
| Elevation + Ruggedness | 4 | 242.1 | 3.15 | 0.087 |
| Age + Elevation | 4 | 242.3 | 3.33 | 0.080 |
| Age*DistRoads + Age + DistRoads | 5 | 242.5 | 3.52 | 0.072 |
| Age*Elevation + Age + Elevation | 5 | 242.8 | 3.87 | 0.061 |
| Age + DistRoads + Elevation | 5 | 243.7 | 4.71 | 0.040 |
| Age + DistTrees + DistPriv | 5 | 243.8 | 4.79 | 0.038 |
| Age*DistTrees + Age + DistTrees | 5 | 243.8 | 4.81 | 0.038 |
| DistPriv + DistTrees | 4 | 244.3 | 5.33 | 0.029 |
| Age*DistPriv + Age + DistPriv | 5 | 244.4 | 5.38 | 0.029 |
| Age + Elevation + DistPriv | 5 | 244.4 | 5.38 | 0.029 |
| Slope + Aspect + Ruggedness + DistTrees | 6 | 244.6 | 5.61 | 0.025 |
| Elevation + DistTrees + Ruggedness + DistRoads | 6 | 244.9 | 5.95 | 0.021 |
| Null | 1 | 295.1 | 56.16 | 0.000 |

We included Animal ID as a random effect in every model. Models with greater than two percent of the cumulative weight are listed below. Top model included
elevation, distance to trees, and an interaction term, accounting for 42.1% of the total weight. Models carrying at least 2% of the total weight were included in the table.

**Table 6. Output from top model (based on AICc) of survival of elk based on based on overall location of the home range on the landscape.**

|  | Estimate | Std. Error | p–Value |
| --- | --- | --- | --- |
| Intercept | 10.784 | 1.447 | < 0.001 |
| Elevation | 0.430 | 0.811 | 0.596 |
| DistTrees | 0.507 | 0.804 | 0.528 |
| Elevation*DistTrees | 1.228 | 0.560 | 0.028 |

about resource selection during the hunting season, however, less research has focused on harvest vulnerability and such studies typically examine risk based on use within the home range, while our study compared vulnerability based on habitat use within home ranges and on the overall location of the home range. Further, our study benefitted from a large sample size and repetition across multiple years. However, some limitations should be taken into consideration as well. Similar studies have incorporated some measure of hunter density or hunter effort [51], which was not available in our study areas. Other habitat variables, such as topographical

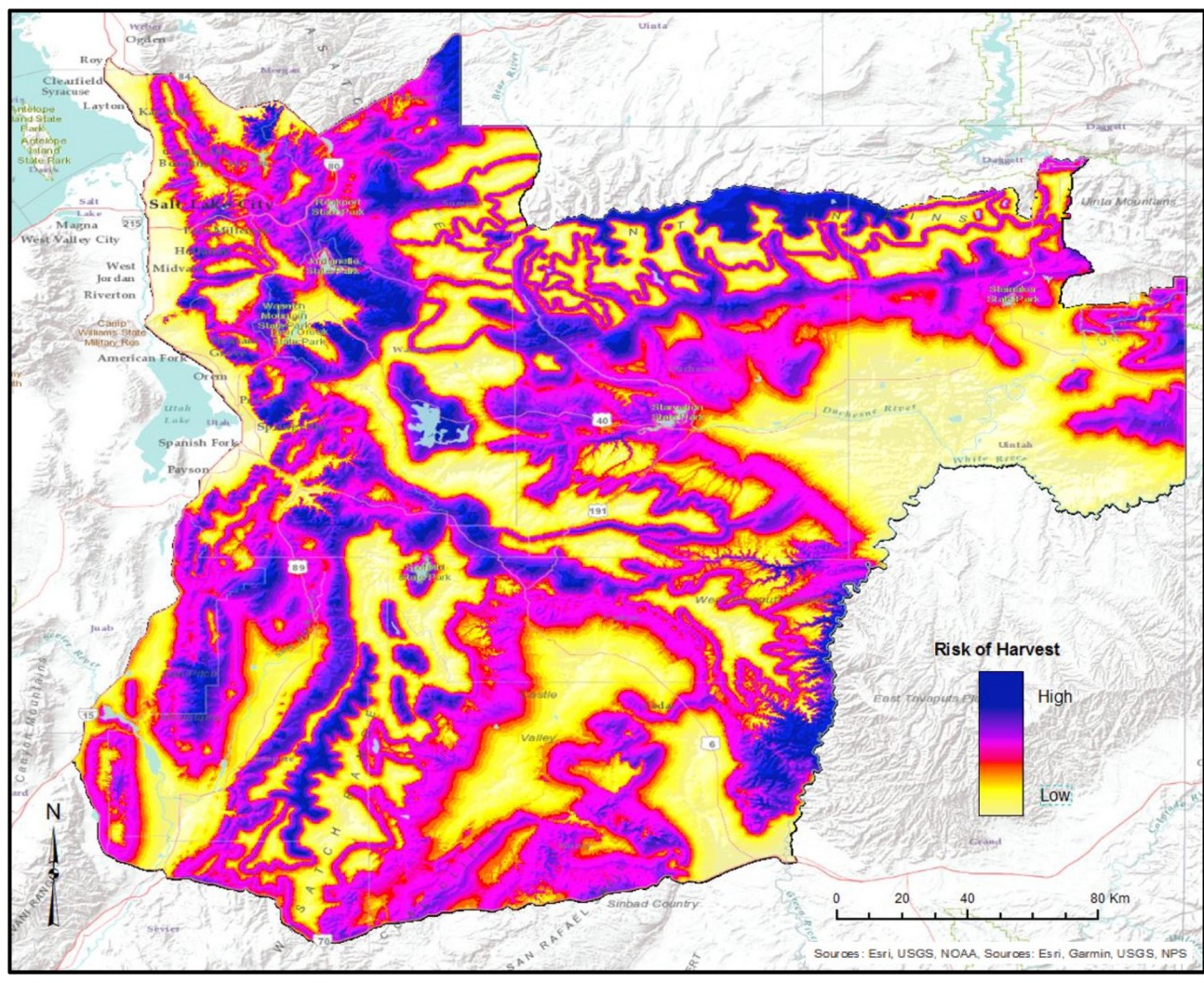

**Fig 5. Heat map of harvest vulnerability of elk based on the location of the home range on the landscape, modeled as a function of elevation, distance to trees, and an interaction between elevation and distance to trees.**

complexity, that were not measured may have also been influential in predicting vulnerability to harvest. Seasonal effects may have influenced the habitat selection of elk, as well. We restricted our sampling period to the hunting season to remove some seasonal variation, however, the hunting season does coincide with a shift from autumn to winter and therefore, changing temperature and weather patterns may influence habitat use of elk. Further, some variation may have existed throughout the duration of the season, between years or between weapon types, however, there is generally a consistent presence of hunters and an increased risk of mortality once the hunting season commences until the close of the season.

Our study supports the idea that elk select for areas with limited hunter access and highlights habitat characteristics that best predict harvest risk of elk in central Utah. These results provide further insight into the responses of game species to hunting pressure and can be used to inform future management policies. By better understanding vulnerability of elk across the landscape, allocation of hunting permits can be adjusted accordingly in areas of high or low vulnerability to better meet population objectives.

## Supporting information

**S1 Table. Correlation values for variables used in resource selection models of cow elk in central Utah during the hunting season, calculated using 'cor()' function in program R.** (DOCX)

**S2 Table.** (CSV)

## Acknowledgments

We are very grateful to the Rocky Mountain Elk Foundation, Sportsmen for Fish and Wildlife, Utah Division of Wildlife Resources, and Brigham Young University for their collaborative efforts. We also thankfully acknowledge Heliwild for their assistance on the wildlife captures and all of our field technicians who assisted on the project. We would also like to graciously acknowledge the editors and reviewers whom provided many helpful comments to improve the manuscript.

## Author Contributions

**Formal analysis:** Maksim Sergeyev.

**Funding acquisition:** Brock R. McMillan, Kent R. Hersey, Randy T. Larsen.

**Methodology:** Brock R. McMillan.

**Project administration:** Brock R. McMillan, Kent R. Hersey.

**Supervision:** Kent R. Hersey, Randy T. Larsen.

**Writing – original draft:** Maksim Sergeyev.

**Writing – review & editing:** Brock R. McMillan, Randy T. Larsen.

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
