## [Decision Letter · Decision Letter 0]

20 Dec 2019

PONE-D-19-22105

Habitat Use and Harvest Vulnerability of Elk (Cervus canadensis): Do Elk Learn to Avoid Hunters as They Age?

PLOS ONE

Dear Mr Sergeyev,

Thank you for submitting your manuscript to PLOS ONE. After careful consideration, we feel that it has merit but does not fully meet PLOS ONE’s publication criteria as it currently stands. Therefore, we invite you to submit a revised version of the manuscript that addresses the points raised during the review process.

As you will see, both Reviewers see clear merits in your contribution, but both raise the point that sex should be better taken into account, either by performing separate analyses or by including sex in your models. Also, Reviewer 1 provided a thorough list of comments that will be very useful to make your manuscript easier to read by non-specialists.

We would appreciate receiving your revised manuscript by Feb 02 2020 11:59PM. To enhance the reproducibility of your results, we recommend that if applicable you deposit your laboratory protocols in protocols.io, where a protocol can be assigned its own identifier (DOI) such that it can be cited independently in the future. For instructions see: http://journals.plos.org/plosone/s/submission-guidelines#loc-laboratory-protocols

We look forward to receiving your revised manuscript.

Kind regards,

Louis-Felix Bersier, Ph.D.

Academic Editor

PLOS ONE

Additional Editor Comments:

You will find my comments on your ms in the attached pdf document (PONE-D-19-22105_LFB.pdf)

Journal Requirements:

2. In your Methods section, please provide additional information on the blood collection and ensure you have included details on (1) locations of where the blood was collected, and (2) the volume of collected blood from each animal.

3. We note that Figures 1 and 5 in your submission contain map/satelliteimages which may be copyrighted.

a. You may seek permission from the original copyright holder of Figures 1 and 5 to publish the content specifically under the CC BY 4.0 license. 

4. Thank you for including your ethics statement:

"Capturing and handling of animals was conducted in accordance with Brigham Young University Institutional Animal Care and Use Committee permit #150112. Animals were not anesthetized or euthanized."

a. Please amend your current ethics statement to confirm that your named ethics committee specifically approved this study.

For additional information about PLOS ONE submissions requirements for animal ethics, please refer to http://journals.plos.org/plosone/s/submission-guidelines#loc-animal-research.

6. Please include a separate caption for each figure in your manuscript.

Reviewers' comments:

Reviewer's Responses to Questions

**Comments to the Author**

1. Is the manuscript technically sound, and do the data support the conclusions?

Reviewer #1: Partly

Reviewer #2: Partly

2. Has the statistical analysis been performed appropriately and rigorously? 

Reviewer #1: No

Reviewer #2: Yes

3. Have the authors made all data underlying the findings in their manuscript fully available?

Reviewer #1: No

Reviewer #2: Yes

4. Is the manuscript presented in an intelligible fashion and written in standard English?

Reviewer #1: Yes

Reviewer #2: Yes

5. Review Comments to the Author

**Reviewer #1**: General comments

Deer populations are traditionally managed by means of hunting. Deer may respond to hunting by using refuges, avoiding human activity centers, modifying movement, activity and habitat selection. Accordingly, knowledge of the effects of hunting on deer behavior is necessary to enhance the effectiveness of deer management. In the present manuscript you determine the habitat characteristics most influential in predicting harvest risk of elk and if elk learned to avoid hunters with age. Although I do not call the relevance of these data into question, I have several points that need to be clarified or refined but see my general and specific comments.

Methods

PLOS is an international journal and not all readers are familiar with elk hunting in the USA. In this regard, you should include additional information in Methods (see points below):

How are elk hunted in the Wasatch Mountains and surroundings? Are there different types of hunting (e.g. hunting with beaters, hunting from a hide…)? Please specify

Could you please provide further information about how elk hunting is organized and if there is a shooting plan (number of males, females and fawns)? If there is a shooting plan you should add it in Methods.

All this information will be useful for the interpretation of your results and for the discussion.

Could you please provide more information about what is exactly meant by private land and why you think that this variable should be included to model elk habitat use and risk of harvest? I am especially wondering why private land should be less accessible to hunters given that some private landowners could be hunters too or allow hunters to hunt on their ground according to my understanding. Clarify please.

Study design and statistical analyses:

You wrote “We analyzed selection preferences during the hunting season using a resource selection function”. For me it is not clear what you aim for by doing so. The chosen approach does not enable to draw any conclusions regarding the hunting effect as you cannot be sure that elk alter habitat selection during hunting hours because of hunting or because of other confounding factors (see below).

If all elk included in the present study are exposed to hunting pressure, it precludes a comparison of hunted with non-hunted individuals in order to detect hunting effects.

Comparing habitat selection during the hunting season to the period before or after is not ideal as seasonal trends in selection and changes in behavior due to hunting can be confounded.

To disentangle other effects (e.g. human disturbances, predators, competition with domestic animals…) from hunting effects, you need to choose a different approach. You could e.g. follow the method described in Behr et al. (2017) and build two separate habitat selection models using two nested data sets: the full data set which includes all location data over the entire year (all data model) and the reduced data set in which you exclude the hunting period from the data and interpolate elk habitat use during the missing hunting period (no-hunting model). As stated by Behr et al. (2017) this method could fail to disentangle a seasonal from a hunting effect if the seasonal driver of habitat selection would perfectly coincide with the hunting season. Hence you need to check if there is a correspondence between the natural history of elk and the timing of the hunting season. If there is no correspondence, you are on the save side.

I recommend to take gender into account in the analyses as vulnerability to harvest based on habitat use could differ between males and females.

You should provide more information on how elk were aged and on how many age categories were considered in the analysis.

I recommend to screen for collinearity between predictors prior to the analysis. Check please (see also my specific comments).

I recommend to perform some validations e.g. Spearman rank correlation from the k-fold cross-validation.

Formatting of the manuscript:

You should ease the work of the reviewers and include line and page numbers.

Data availability:

In contrary to the PLOS Data policy, you did not make all data underlying the findings in our manuscript fully available.

Specific comments

Major comments

Introduction

Page 4: Other aspects such as vigilance, site familiarity could influence the risk of harvest as well. I suggest adding them here. Risk of harvest does not only depend on hunter accessibility other factors could be important as well. E.g. once a deer is spotted/discovered the probability to be harvested could depend on sex, antler size, body condition. It depends furthermore on the type of hunting and on the hunting plan (see above). Clarify please.

Page 4: Besides hunting other factors could influence elk habitat selection including disturbances, interspecific competition (e.g. competition with domestic animals) and predators. In these regards, how can you be sure that the observed difference in habitat selection between night and day is solely the effect of hunting and not due to other confounding factors? Clarify please.

Could you, please provide more information about the domestic livestock (page 5: Methods/Study area:)? Which species, herding practices (are they grazing in fenced pastures), is livestock staying out at night, etc.? Could they potentially compete with elk? Clarify please and discuss.

Are predators such as cougar and wolves present in your study area (these species are manly nocturnal) and could they have a confounding effect with hunting? Clarify please and discuss.

Even if you could exclude livestock and predators there could be other confounding factors you do not know about and hence it could be worthwhile to use an approach that enable to disentangle hunting from other effects (see my suggestion above).

Methods

Page 6: “…and age based on dental wear”. Knowing that age estimation by means of dental wear are not very accurate, I was wondering, which age categories were considered in the present study? Clarify please.

Page 6: “minimum convex polygon” Why did you chose MCP knowing that these can include large proportion of unsuitable habitat especially in a human dominated landscape? I would have expected Kernel home ranges. Please justify why you used MCP in the present study.

Analysis

This is section is not well presented and needs some reordering. I recommend adding subtitles before each section. According to the information provided in the manuscript, I see three topics/sections:

1) Habitat selection to examine habitat use during the hunting season by taking into account differences in selection between day and night. Here it would be worthwhile to choose an approach that enables to disentangle hunting from other confounding effects (see my suggestion above).

2) Evaluation of vulnerability to harvest within the home range including learned hunter-avoidance by older elk.

3) Evaluation of vulnerability to harvest based on the overall location of home range on the landscape including learned hunter-avoidance by older elk.

Page 6: “Based on known locations of use…Lele and Keim, 2006). Please provide further details regarding the sampling design used in the present study fort both within home range and overall location of home range on the landscape. For example, how did you asses availability (number of random locations), etc…See for example Hebblewhite et al. (2005).

Page 6 to page 7: “To examine differences in selection between day …and night…” how did you define day and night in your study? Did you consider that days length changes over the course of the study? Clarify please and provide this information in the manuscript.

Page 7: “We modeled risk of harvest by hunters using logistic regression with 1 corresponding to survival and 0 to harvest…”

In case if elks were harvested during different type of hunting in the present study, I think that it would be wise to consider different categories in the analyses. Clarify please.

I suspect that the harvest risk of a deer varies over the course of the day (highest probability just before dusk and after dawn) and study period (more deer are shot at the beginning of the hunting period then towards the end) and depends as well on the characteristics (sex, age, condition) of each individual. You did not consider these aspects in the present manuscript.

In these regards, could you please provide data about the number of deer (males and females) that are present in your study area, the proportion of collard individuals (males and females) and the proportion of individuals, present in the study area, which are shot each year.

Among the harvested individuals could you please provide information about the sex and age.

Could you please provide further information about the distribution of the number of deer shot over the daily cycle and study period in a graph.

If you have access to the locations of the deer that are shot each year you could use these data to validate your heat map of elk harvest vulnerability.

Page 7: “We included variables for distance to roads, aspect, elevation, slope, terrain ruggedness, distance to tree cover and distance to private land.”

Could you please provide further details regarding the source of the variables used in the present study and their resolution, ideally in a table?

Could you please explain how these variables were sampled in the GIS?

Please provide the size of the radius around each location for the habitat use within home range and the size of the radius around the centroid of each home range for the habitat use at a broader scale based on the overall location of the home range on the landscape.

I do not understand why you used the centroid and did not calculate e.g. the average elevation within each MCP? Clarify please.

Which road categories were considered in the analysis (main roads, forest roads, hiking trails…)? Clarify please.

How did you calculate aspect? Did you use it as a categorical value in the analysis? Clarify please.

Page 7: “We evaluated vulnerability to harvest based on use within the home range by averaging data from all locations within the home range and considered each hunting season from every elk as an individual observation (Hayes et al. 2002).” Not too clear, see also my comment regarding the sampling design used in the present study. Clarify please.

Results

Page 8: How many females and males did you collars?

Could you please summarize the number of females and males collared per age category in a table (elk ID, sex, age, date of capture, date of loss, reason for loss, survey period (number of days), number of fixes)?

Do MCP sizes differ between males and females? If yes, you should take this into consideration in the analysis.

Why did you pool the sexes in the analyses?

I suspect that harvest vulnerability based on habitat varies between sexes (males are more vulnerable than females because of their antlers). Hence, I recommend to take gender into account in the analyses. Do you know if females had calves or not? If yes, you could consider this in the analyses as well. Clarify please.

Page 8: “Steep slopes and rugged terrain were correlated with decreased use.“ Why did you not screen for collinearity between predictors prior to the analysis? Clarify please.

Page 9: “We restricted the model set to locations collected during hunting hours (30 minutes prior to sunrise – 30 minutes past sunset) as animals were at no risk of harvest outside this period.” This is not a result and should be moved to the appropriate section in Methods.

Why did you consider a duration of 30 minutes any justification? Clarify please.

Page 9: “The top model included an interaction between elevation and distance to trees (p = 0.028, Table 6) suggesting that at higher elevations, distance to trees became more influential in predicting harvest risk.” Could you please provide further information on how (direction) harvest risk varied with elevation and distance to trees? Please add this information in the manuscript.

Page 9: “As we were unable to model age across the landscape, the top model based on home range characteristics was used to create a heatmap of harvest vulnerability across the study area (Figure 5) to illustrate high-risk areas.” In contrary to the habitat selection within home ranges you did not provide any result here regarding learned hunter-avoidance by older elk at the broader scale, is there a reason? Clarify please.

Discussion

Page 11: “This was likely due to public land generally occurring at higher elevations than private land within our study area; as hunting primarily occurred on public land, this may explain the decreased survival at higher elevations.” In this regard you should have screened for collinearity between predictors prior to the analysis.

Page 11: “Mature bull elk in Michigan had greater flight distances than yearling bulls, in a population where mature bulls were harvested at five times the rate of yearling bulls (Bender et al. 1999).” This is not really connected to the topic of the present manuscript. This is a behavioral difference and has nothing to do with habitat selection and hence should be removed from the discussion.

Page 12: “Additional work may show patterns of hunter avoidance by elk in central Utah…” Not clear what you mean here. Clarify or remove please.

Page 12: “Based on the centroid of the home range, risk of harvest was best predicted by distance to trees, elevation, and an interaction between the two. The interaction term was positive, suggesting that at higher elevations, survival was higher with increasing distance to trees…” This belongs to results and hence should be moved to this section. Adjust please.

Page 13: “These results can provide further insight into the responses of game species to hunting pressure and can be used to inform future management policies.” This statement is too general. Here you should make some clear and concrete recommendations on how your results can be used to inform management policies. According to your findings what adjustments would be needed in the current management policies? Adjust please.

Minor comments

Introduction

Page 2: “Selection of habitats may be influenced by …” you could add inter- and intraspecific competition as well here.

Methods

Page 6: “late August through January 31st” Could you please provide the exact period?

Results

Page 8: “We evaluated habitat selection in the context of harvest vulnerability within home ranges and on a broader scale to evaluate position of home range on the landscape. We evaluated harvest risk at two scales in order to determine vulnerability based on use within an animals home range as well as based on the overall location of the home range on the broader landscape.” Both sentences have the same meaning. Moreover, this section belongs to Methods and hence should be deleted as these aspects were already mentioned in the corresponding section.

Page 8: “Overall, elk selected…” Not clear what you mean by overall and to which results (e.g. Table) you are referring to here? Clarify please.

Page 9: I suggest changing “Additionally, survival was lower at higher elevations (Figure 3) and for older animals (Figure 4).” to “Additionally, harvest risk was higher at higher elevations (Figure 3) and for older animals (Figure 4).” Because you specifically looked at harvest risk and not overall survival which could include other mortality risk as well. This comment is valid for the whole manuscript. Please check throughout the manuscript and adjust where necessary.

Table 1, legend: “AICc model selection results for 27 candidate models of habitat use.” Only five models are provided here. What is meant by “Day”? Clarify and adjust please.

Table 2, legend: What is meant by Day? Clarify please.

Table 3, legend: “AICc model selection results for 20 candidate models of survival based on habitat use.” Only six models are provided here. Clarify and adjust please.

Table 5, legend: “AICc model selection results for 20 candidate models of survival based on overall location of the home range on the landscape. Only 14 models are provided here. What is meant by “Day”? Clarify and adjust please.

Figure 1: What is the additional information provided by the inset? Instead I suggest showing the location of the study area in the USA. Adjust please.

Reference cited in this review not already cited in the present manuscript

Gehr, B., E. J. Hofer, M. Pewsner, A. Ryser, E. Vimercati, K. Vogt, and L. F. Keller. 2017. Hunting-mediated predator facilitation and superadditive mortality in a European ungulate. Ecology and Evolution 8:109–119.

**Reviewer #2**: Very interesting paper, well written and the statistical analyses seems up to date. One thing that comes to mind when looking at your study is that other studies seem to account for the sex of the animal. I am not familiar with the exact policies in Utah, but my experience from elk hunting suggests that hunters have different attitudes towards hunting females and males, often looking for trophies in males, and meat in females (and spikes). Trophies come at a relative high age, also proportion of tags may differ compared to what is available for each sex. I am not sure it is possible, but it would be nice to see if it is possible to distinguish differences if the analyses were re run, once with males and once with females. A study from Alberta by Ciuti et al. (2012), together with the paper by Thurjell et. al that you already cite suggests that, at least in Alberta, the high hunting pressure on males leads more to a selection of traits than a learning experience. There is also the issue of how gregarious animals are during hunting season, it is hard to imagine a learning process without members of a group being shot. As males are less gregarious than females, we would expect a different process there (even though effects has been found in bulls in Michigan as you state).

There are also other studies that suggest lower mortality with age (Wright 2006), thus it would be interesting if you made some attempt to dive in to the differences compared to those studies. (By more detailed analysis as suggested, or at least by a discussion on the subject).

As for the hunting season, from late August to Jan 31st is quite a long time, with quite different weather conditions for the elk (And different weapons and tactics used by hunters, as Utah has an archery season and a Muzzel loader season). Yet I see no attempt to account for that, there are several ways to do this. If you want to account for seasonality and still have the effect of hunting (as they will be mixed up), one way of doing this is to add a factor weekend or weekdays, as we could expect hunting pressure is generally higher on Saturdays and Sundays.

As for minor details such as language and grammar, as a non native English speaker, that is not my forte, and I tried to focus on the structure here.

Ciuti S, Muhly TB, Paton DG, McDevitt AD, Musiani M, Boyce MS. Human selection of elk behavioural traits in a landscape of fear. Proc R Soc B-Biol Sci. 2012;279(1746):4407–16.

Wright GJ, Peterson RO, Smith DW, Lemke TO. Selection of northern Yellowstone elk by gray wolves and hunters. J Wildlife Manage. 2006;70(4):1070–8.

6. PLOS authors have the option to publish the peer review history of their article (what does this mean?). If published, this will include your full peer review and any attached files.

Reviewer #1: Yes: Fridolin Zimmermann

Reviewer #2: No

---

## [Author Response · Author response to Decision Letter 0]

13 Mar 2020

Thank you for the revisions provided to our submitted manuscript. We incorporate the recommended changes and have provided an updated draft of the manuscript. The specific edits that were made are detailed in the "Response to Reviewers" document included in the resubmission. Thank you again for your time and consideration.

---

## [Decision Letter · Decision Letter 1]

21 May 2020

PONE-D-19-22105R1

Habitat Use and Harvest Vulnerability of Elk (Cervus canadensis): Do Elk Learn to Avoid Hunters as They Age?

PLOS ONE

Dear Dr. Sergeyev,

Thank you for submitting your manuscript to PLOS ONE. After careful consideration, we feel that it has merit but does not fully meet PLOS ONE’s publication criteria as it currently stands. Therefore, we invite you to submit a revised version of the manuscript that addresses the points raised during the review process.

As you can see, both reviewers were pleased to see the revisions and improvements to your manuscript. For submission of a revised manuscript, please provide additional justification for the inclusion or exclusion of classes of animals as described by Reviewer 1. Additionally, note Reviewer 1's comments about confounding factors in your analysis. Reviewer 2 provides some suggestions for improving the flow and structure of the manuscript. In addition, please note their request for additional discussion of why female elk did not show evidence of some learning, as has been previously suggested.

We look forward to receiving your revised manuscript.

Kind regards,

Christopher James Johnson, Ph.D.

Academic Editor

PLOS ONE

Reviewers' comments:

Reviewer's Responses to Questions

**Comments to the Author**

1. If the authors have adequately addressed your comments raised in a previous round of review and you feel that this manuscript is now acceptable for publication, you may indicate that here to bypass the “Comments to the Author” section, enter your conflict of interest statement in the “Confidential to Editor” section, and submit your "Accept" recommendation.

Reviewer #1: (No Response)

Reviewer #2: (No Response)

2. Is the manuscript technically sound, and do the data support the conclusions?

Reviewer #1: No

Reviewer #2: Partly

3. Has the statistical analysis been performed appropriately and rigorously? 

Reviewer #1: No

Reviewer #2: Yes

4. Have the authors made all data underlying the findings in their manuscript fully available?

Reviewer #1: Yes

Reviewer #2: Yes

5. Is the manuscript presented in an intelligible fashion and written in standard English?

Reviewer #1: Yes

Reviewer #2: Yes

6. Review Comments to the Author

Reviewer #1: General comments

Although the manuscript has been improved. There are still several points that need further attention. Especially the manuscript is technical not sound as there are several problems in your sampling design and statistical analyses that preclude making sound conclusions at the current stage. But see my general and specific comments below.

In this analysis you apparently only included animals lost to hunter harvest (as you have written in your answers to reviewers).Could you please justify (rational behind) why you only used individuals lost to hunter harvest? Beside that this is not clearly stated in the material and method, I think that you should include all elk cows followed by means of GPS telemetry over the course of this survey in the analyses including those that were not shot over all the hunting seasons, as done by McCorquodale et al. (2003). Check their paper to see exactly how they proceed. Exclusion of animals that survived throughout all hunting seasons might bias our analysis toward elk using less secure environments. Clarify please and adjust if necessary.

In this regard you should summarize the number of cows and bulls collared per age category in a table (elk ID, sex, age, date of capture, date of loss, reason for loss (hunting, predation,…), survey period (number of hunting seasons, number of days, number of fixes). Moreover, you should add in the results how many elk died and how many deaths were hunting related (like McCorquodale et al. 2003).

You mentioned that habitat selection by elk is likely greatly confounded by seasonal effects in Utah. The heavy snows will typically push elk to lower elevations during the colder periods, while availability of forage and extreme heat likely influence habitat use during the summer. For this reason, you restricted our analysis to the hunting season to minimize these seasonal differences. Given that the hunting season stretches from the end of August and is concluded on January 31st, it encompasses the transition from autumn to winter that include significant changes e.g. heavy snows will typically push elk to lower elevations during the colder periods, with snow elks can easier be detected by hunters thanks to the tracks they leave in the snow. Hence as the hunting season stretches over quite a long period, with quite different weather conditions for the elk and different weapons and tactics used by hunters resulting in changes in the hunting success you need to account for that as already recommended by the other reviewer in the analysis.

You excluded animals with less than 50 locations during the hunting season to avoid biased estimates of home ranges. Given that your interest in estimating home ranges was principally to define sampling frames for quantifying characteristics of areas used by elk (rather than a means of estimating a home range as stated in your response to reviewer’s comments), I do not see why you should discard data from elk that were killed before a large sample of relocations were obtained. Exclusion of such animals might bias your analysis toward elk using higher-security environments. Clarify and discuss please.

Variables were screened for collinearity in program R (line 207). Which analysis did use to test for multicollinearity between predictors and what was the threshold? Please provide the results of this analysis as a supplementary material.

Could you please provide more details on how you proceeded with the validation (lines 214-215)? How many folds did consider? Clarify please.

Furthermore, I could not find any values of the Spearman’s rank correlation in Table 1 (lines 260-261). What is the value of the Spearman’s rank correlation? Clarify please.

Why do you just do a validation for the “habitat use during hunting season” but not for the remaining two analyses “harvest vulnerability based on 3rd order habitat use” and “harvest risk based on location of home range”. Clarify please.

As your study focus solely on elk cows you need to adjust the title and abstract and the main text accordingly. In the discussion you should focus mainly on findings from other studies related to elk cows, especially if differences between bulls and cows are expected.

You indicated that wolves and bears are absent from the area, however, cougars are present and stated that predation of elk by cougars is generally minimal and typically cougars target young or old individuals in poorer condition. Even though predation of elk by cougars is minimal, even low carnivore densities affect prey behaviour (Kuijper et al. 2016). Besides, I question that cougars target old individuals in poorer condition. I would have rather expected to see such a pattern with wolves. Cougars are stalk and ambush predators and hence have a higher probability to catch less alert prey and exert selection pressure rather on the behavior of their prey (less alert prey have a higher chance to be predated) while wolves, because of the way they hunt, exert selection pressure rather on the condition of their prey. If you were not able to take such confounding effects into account in the analyses you should at least discuss this shortcoming in the discussion of the manuscript.

You wrote that you examined whether habitat use changes in regard to these variables when hunters are present versus absent. Yes, but what if in your study area period of times when hunters are present versus absent overlap with other factors such as disturbances and predators. In this case you cannot distinguish hunting from these confounding effects. You should at least discuss this shortcoming in the discussion of the manuscript.

No track changes are visible in the document “Revised Manuscript with Track Changes”. Please make sure that you upload the right files to ease the work of the reviewers.

Specific comments

Lines 121-123: Could you please specify roughly the period in months of the archery only, muzzleloader only, and any weapon hunting seasons?

Lines 173-175: Did you calculate an MCP for each hunting season that an elk lived/was followed by means of GPS telemetry? Clarify please

Lines 190-192: This sentence can be deleted as this information appears earlier.

Lines 198-203: In the last sentence you repeat what you have already written in the first sentence. Shorten please

Lines 223-226: You repeat what you have written in the previous subchapter. Hence this part can be shortened considerably.

Lines 306-312: Here you repeat the results provided at line 300-302. The whole section starting from line 300 to 312 should be substantially simplified and streamlined. Please rewrite it.

Line 322: “Survival decreased with increasing elevation.” Could also be related to snow cover!? Clarify please and discuss if deemed important.

Lines 336-339: This sentence is not really connected to the topic of the present manuscript. This is a behavioral difference and has nothing to do with habitat selection. Moreover, you restricted your analyses to elk cows. Hence this sentence should be removed from the discussion.

Lines 343-345: This sentence can be removed as it says nothing substantial

Lines 372-374: It is not clear what you mean exactly here. Could you please further develop this idea. In other words, how would you proceed to allocate hunting permits?

Reference cited in my review

Kuijper D P J,Sahlen E,Elmhagen B,Chamaille-Jammes S,Sand H,Lone K, et al. Pawswithoutclaws? Ecological effects of largecarnivores in anthropogenic landscapes. Proceedings of the Royal Society B: Biological Sciences 2016; 283: 20161625. https://doi.org/10.1098/rspb.2016.1625 PMID: 27798302

Reviewer #2: Now that it is clear that this is not an overall elk-study, I think the title and onwards should be clearly stated as female Elk instead of just elk. This follows through the manuscript.

Line 167-172 is a justification for methods used, this is more fitting in the introduction, it also suggests that there is literature on game species about awareness of risk etc… this needs to be referenced directly after the statement.

177-180, do not mention concerns and how you feel in the materials and methods section, describe which method you choose and why, that’s it.

234 Was the modelled random effect regarding slope or intercept or both, and why did you select that way? Compare to Thurfjell 2017 where the formulation of the random effect was used to test if there was a learning effect.

255 How often were locations taken? What was the success rate?

In the discussion, I lack some hypothesis as to why female elk did not show learning, as opposed to precious studies, also why do you think they are more vulnerable at older ages? Think a bit about theese results, and try and formulate something that can be useful for further studies.

7. PLOS authors have the option to publish the peer review history of their article (what does this mean?). If published, this will include your full peer review and any attached files.

Reviewer #1: Yes: Fridolin Zimmermann

Reviewer #2: Yes: Henrik Thurfjell

---

## [Author Response · Author response to Decision Letter 1]

16 Sep 2020

We have uploaded a cover letter detailing the specific edits made in response to the comments provided. We are very grateful to the editors and reviewers for their time in reviewing our study and believe that the comments provided have improved the final product greatly.

---

## [Decision Letter · Decision Letter 2]

27 Oct 2020

PONE-D-19-22105R2

The Influence of Habitat Use on Harvest Vulnerability of Cow Elk (Cervus canadensis)

PLOS ONE

Dear Dr. Sergeyev,

Thank you for submitting your manuscript to PLOS ONE. After careful consideration, we feel that it has merit but requires some minor revisions to meet PLOS ONE’s publication criteria. Therefore, we invite you to submit a revised version of the manuscript that addresses the points raised during the review process.

Most importantly, I recommend revising the section of the Discussion that was indicated in the review for clarity and conciseness. Please note that the line numbers correlate with the "Track Changes" version of the submitted revision.

We look forward to receiving your revised manuscript.

Kind regards,

Christopher James Johnson, Ph.D.

Academic Editor

PLOS ONE

Reviewers' comments:

Reviewer's Responses to Questions

**Comments to the Author**

1. If the authors have adequately addressed your comments raised in a previous round of review and you feel that this manuscript is now acceptable for publication, you may indicate that here to bypass the “Comments to the Author” section, enter your conflict of interest statement in the “Confidential to Editor” section, and submit your "Accept" recommendation.

Reviewer #1: All comments have been addressed

2. Is the manuscript technically sound, and do the data support the conclusions?

Reviewer #1: Yes

3. Has the statistical analysis been performed appropriately and rigorously? 

Reviewer #1: Yes

4. Have the authors made all data underlying the findings in their manuscript fully available?

Reviewer #1: Yes

5. Is the manuscript presented in an intelligible fashion and written in standard English?

Reviewer #1: Yes

6. Review Comments to the Author

Reviewer #1: Line numbers correspond to the document where the track changes are still visible

The manuscript has been considerable improved, and I have only a few minor comments

Best wishes

Fridolin Zimmermann

Line 22: change "Elk" to "female Elk"

Line 30: change "Elk" to "female Elk"

Lines 362-363: You did not study Elk activity. Please adjust the sentence accordingly

Lines 378-379: I do not see the rational given that harvest vulnerability is positively correlated with elevation. Alternatively, snow may have increased elk detectability by hunters at higher elevation and thereby increased the likelihood of mortality at higher elevation. Clarify please.

Lines 383-392: Please reformulate this section in a more concise manner.

Finally, I recommend you to acknowledge reviewers. I consider this good habit as one of the most important point of academic ethics.

7. PLOS authors have the option to publish the peer review history of their article (what does this mean?). If published, this will include your full peer review and any attached files.

Reviewer #1: **Yes: **Fridolin Zimmermann

---

## [Author Response · Author response to Decision Letter 2]

8 Nov 2020

We are grateful to the editor and reviewers for considering our manuscript. The specific edits are further clarified in the response to reviewers document. Thank you again for your time and consideration

---

## [Editor Report · Decision Letter 3]

11 Nov 2020

The Influence of Habitat Use on Harvest Vulnerability of Cow Elk (Cervus canadensis)

PONE-D-19-22105R3

Dear Dr. Sergeyev,

We’re pleased to inform you that your manuscript has been judged scientifically suitable for publication and will be formally accepted for publication once it meets all outstanding technical requirements.

Kind regards,

Christopher James Johnson, Ph.D.

Section Editor

PLOS ONE
---

## [Editor Report · Acceptance letter]

13 Nov 2020

PONE-D-19-22105R3 

The Influence of Habitat Use on Harvest Vulnerability of Cow Elk (*Cervus canadensis*) 

Dear Dr. Sergeyev:

I'm pleased to inform you that your manuscript has been deemed suitable for publication in PLOS ONE. Congratulations! Your manuscript is now with our production department. 

Kind regards, 

on behalf of

Dr. Christopher James Johnson 

Section Editor

PLOS ONE